# A Novel PDK1/MEK Dual Inhibitor Induces Cytoprotective Autophagy via the PDK1/Akt Signaling Pathway in Non-Small Cell Lung Cancer

**DOI:** 10.3390/ph16020244

**Published:** 2023-02-06

**Authors:** Rangru Liu, Zhuo Chen, Gaoyun Hu, Zutao Yu, Qianbin Li, Danqi Liu, Ling Li, Zhaoqian Liu

**Affiliations:** 1Department of Clinical Pharmacology, National Clinical Research Center for Geriatric Disorders, Hunan Key Laboratory of Pharmacogenetics, Xiangya Hospital, Central South University, Changsha 410008, China; 2Key Laboratory of Tropical Translational Medicine of Ministry of Education, NHC Key Laboratory of Control of Tropical Disease, School of Pharmacy, Hainan Medical University, Haikou 570100, China; 3Department of Medicinal Chemistry, Xiangya School of Pharmaceutical Sciences, Central South University, Changsha 410013, China; 4Department of Pharmacy, Xiangya Hospital, Central South University, Changsha 410008, China; 5Department of Pharmacy, The Fifth Affiliated Hospital, Sun Yat-sen University, Zhuhai 519000, China

**Keywords:** compound YZT, a PDK1/MEK dual inhibitor, the PDK1/Akt pathway, autophagy, apoptosis

## Abstract

In a preliminary study, we synthesized a series of new PDK1/MEK dual inhibitors. Antitumor activity screening showed that Compound YZT exerts a strong inhibitory action in A549 cells. However, the specific mechanism of YZT against non-small cell lung cancer (NSCLC) is largely unknown. This work confirmed the anti-proliferation and pro-apoptosis effects of YZT in NSCLC cells. Furthermore, YZT promotes autophagy and provokes complete autophagic flux in NSCLC cells. Notably, compared with YZT alone, the combination of YZT with the autophagy inhibitor chloroquine (CQ) or 3-methyladenine (3-MA) markedly strengthened the anti-proliferative and pro-apoptotic actions, suggesting that YZT-induced autophagy is cytoprotective. We further found that YZT-induced autophagy may exert a cytoprotective function by preserving the integrity of mitochondria and decreasing mitochondrial apoptosis. Moreover, Kyoto Encyclopedia of Genes and Genomes (KEGG) analysis suggested that PDK1 is an upstream protein of the Akt/mTOR axis and western blotting verified that YZT induces autophagy by the PDK1/Akt/mTOR signaling axis. Finally, YZT plus CQ significantly enhanced the anticancer activities compared to YZT alone in an animal study and immunohistochemistry showed that the level of LC3 was increased by YZT, which is in line with the in vitro results. In short, our study provides reliable experimental basis for developing Compound YZT as a new chemotherapeutic drug candidate and suggests that combined administration of YZT with CQ is a potential therapy against NSCLC.

## 1. Introduction

Lung carcinoma ranks at the top of tumor-related deaths worldwide [1]. Non-small cell lung cancer (NSCLC) is the most prevalent category and accounts for 84% of lung carcinomas. NSCLC treatment mainly includes surgery, radiotherapy and chemotherapy. Platinum-based doublet therapy has been the normative first-line treatment for advanced NSCLC patients. Nevertheless, a large quantity of NSCLC patients cannot obtain satisfactory treatment due to the defects of chemotherapeutics, such as poor selectivity, severe side effects, drug resistance and a high recurrence rate [2]. Therefore, it is imperative to develop novel chemotherapeutic medicines for NSCLC.

To date, research on new anticancer drugs has evolved from traditional nonselective cytotoxicity to accurately targeting multiple signaling pathways or proteins. Growing evidence suggests that the dual blockade of MEK and PI3K is an effective strategy for cancer therapy [3,4,5]. The MEK inhibitor PD0325901 possesses an activity-indispensable diphenylamine scaffold to bind in the allosteric site. The PDK1 specific inhibitor BX517 can occupy the same binding site of the adenosine triphosphate (ATP) and form vital Hydrogen bonds with the backone of the hinge region. Herein, we synthesized and built a compound library with new types of MEK/PDK1 dual inhibitors by taking PD0325901 and BX517 as lead compounds according to the fragment-based drug design (FBDD) strategy (Figure 1a). The anticancer activity screening indicated that, among these synthesized compounds, Compound YZT (Figure 1b, chemical name:3-(Benzo[*d*][1,3]dioxol-5-ylmethyl)-3-hydroxy-5-(4-(trifluoromethyl)phenylamino)indolin-2-one) has the strongest inhibitory action in A549 cells [6]. However, the underlying mechanisms of YZT against NSCLC are largely unclear.

Autophagy is a highly regulated recycling process during which intracellular superfluous, damaged or aged organelles and parts of the cytoplasm are sequestered with autophagosomes and delivered into lysosomes for degradation to generate the necessary energy and metabolic precursors [7]. Autophagy can be provoked by many stimuli, such as nutrient deficiency, hypoxia, oxidative stress and drugs, although it also occurs under basic physiological conditions [8]. Drug-induced autophagy is a “double-edged sword” in cancer therapy. It has been reported that autophagy induced by some anticancer compounds or agents is advantageous for the survival of tumor cells [9,10]. In contrast, in some specific circumstances, drug-provoked autophagy itself can lead to autophagic type II cell death [11,12,13]. The AMPK/mTOR and PI3K/AKT/mTOR pathways are two classical signaling pathways that induce autophagy [14]. Considering that YZT is synthesized by targeting the PI3K/PDK1/AKT pathway, we wondered whether YZT promotes autophagy and attempted to further investigate the role of autophagy in its anticancer activity.

This work demonstrated that YZT possesses anti-proliferative and pro-apoptotic effects in NSCLC cells. In addition, YZT promotes autophagy by the PDK1/Akt axis. Interestingly, autophagy inhibition markedly enhances the anticancer activities of YZT. We further proved that autophagy inhibition aggravates YZT-induced mitochondrial apoptosis, thereby illuminating that YZT-induced autophagy may exert a cytoprotective function by preserving the integrity of mitochondria and decreasing mitochondrial apoptosis. Finally, the enhanced anti-NSCLC activities of the combination of YZT with the autophagy inhibitor chloroquine (CQ) were confirmed in nude mice in vivo. In brief, this work clearly demonstrates that YZT is a promising novel type of anti-NSCLC drug candidate and that combined administration of YZT with CQ might be a potential alternative therapy for NSCLC.

## 2. Results

### 2.1. Compound YZT Suppresses NSCLC Cell Viability and Growth

To explore the antitumor activities of YZT in NSCLC cells and its safety to normal lung cells, the influence of YZT was studied on the viability of A549 and H1299 cells and one normal lung cell lineMRC-5. CCK-8 assay showed that YZT reduced NSCLC cell viability in a dose-dependent manner but hardly affected MRC5 cell viability (Figure 1c). BrdU labeling and colony formation assays further confirmed the inhibitory action of YZT on NSCLC cell proliferation (Figure 1d,e). In addition, Annexin V-FITC/PI apoptosis assay indicated that YZT dose-dependently induces apoptosis in NSCLC cells and has almost no impact on MRC-5 cells (Figure 1f). A similar conclusion was obtained from western blotting, which was applied to measure the apoptosis marker proteins active caspase-9, active caspase-3 and cleaved PARP (Figure 1g). Finally, NSCLC cells were pretreated with the pancaspase inhibitor Z-VAD-fmk before YZT treatment and immunoblotting demonstrated that Z-VAD-fmk markedly inhibited YZT-induced activation of caspase-9 and caspase-3 and cleavage of PARP (Figure 1h), suggesting that caspase activation is necessary for inducing apoptosis.

In brief, these data reveal that YZT is cytotoxic and induces NSCLC cell apoptosis in a caspase-dependent way.

### 2.2. Compound YZT Provokes Autophagy in NSCLC Cells

Growing evidence has suggested that autophagy exerts a pivotal function in the anticancer activities of chemotherapeutic drugs [15,16,17]. To clarify whether YZT induces autophagy in NSCLC cells, the classical autophagy protein LC3 was detected by immunoblotting, which showed that YZT increased the level of LC3-II in a dose-dependent manner (Figure 2a). In addition, immunofluorescence indicated that the quantity of LC3 puncta was significantly increased by YZT (Figure 2b). Furthermore, transmission electron microscopy(TEM) was employed to directly observe the morphology of autophagosomes. TEM showed that the cells in the YZT group clearly had a classic ultrastructural morphology of autophagosome bilayer vacuoles compared to the DMSO control (Figure 2c). Finally, another typical characteristic of autophagy, acidic vesicular organelles (AVOs), can be examined by AO staining. The results indicated that YZT facilitated the formation of AVOs in a dose-dependent way (Figure 2d).

All of the data demonstrate that YZT induces autophagy in NSCLC cells.

### 2.3. Compound YZT Induces Intact Autophagic Flux

To delve into the effect of YZT on autophagy, it is worth determining whether YZT promotes complete autophagic flux. For this purpose, firstly, immunoblotting was adopted to measure the autophagy-related proteins SQSTM1, Atg5 and Beclin1. The results indicated that Atg5 and Beclin1 were upregulated in a dose-dependent manner, but SQSTM1 showed no significant difference following YZT administration (Figure 3a). Furthermore, the level of LC3 was tested by immunoblotting to determine the impact of the autophagy inhibitor CQ on YZT. Immunoblotting showed that LC3 was markedly increased by YZT plus CQ compared with YZT alone (Figure 3b). Similar results were obtained when immunofluorescence was used to observe the endogenous LC3 puncta (Figure 3c,d). Because autophagosomes can fuse with lysosomes in complete autophagic flux, immunofluorescence was employed to monitor this process by observing the colocalization of LC3 and LAMP1 which is a biomarker protein of lysosomes. The analysis of the Pearson correlation coefficient revealed that, as the negative control, the autophagy inhibitor CQ could propel the segregation of LC3 and LAMP1and, closely analogous to the autophagy inducer RAPA, YZT promoted the overlap of LC3 and LAMP1 (Figure 3e,f).

Briefly, these results show that YZT promotes complete autophagic flux in NSCLC cells.

### 2.4. Autophagy Blockade Strengthens the Anticancer Activities of Compound YZT

To investigate the effect of autophagy on YZT-mediated proliferation inhibition and proapoptosis, firstly, the autophagy inhibitors CQ and 3-methyladenine (3-MA) at the appropriate concentrations were adopted to suppress YZT-induced autophagy under the precondition that CQ and 3-MA themselves do not affect cell proliferation and promote apoptosis. Subsequently, CCK-8 and BrdU labeling assays were used to study the impact of autophagy blockade on YZT-mediated cytotoxicity. CCK-8 assay revealed that both CQ and 3-MA aggravated the impairment of YZT on cell viability (Figure 4a,b). The same conclusion was drawn from BrdU labeling detection (Figure 4c). 

Next, the impact of autophagy inhibition on YZT-mediated proapoptosis was estimated by Annexin V-FITC/PI apoptosis detection. The results indicated that the combination of YZT with CQ or 3-MA significantly accelerated cell apoptosis compared with YZT alone (Figure 4d,e). A similar conclusion was drawn when western blotting was used to examine the levels of active caspase-9, active caspase-3 and cleaved PARP (Figure 4f,g).

In summary, inhibition of autophagy intensifies YZT-mediated proliferation inhibition and proapoptosis, and YZT-induced autophagy is cytoprotective in NSCLC cells.

### 2.5. Autophagy Inhibition Exacerbates YZT-Mediated Mitochondrial Apoptosis

Considering that our previous study verified that the novel PDK1/MEK dual inhibitor can promote mitochondrial apoptosis in NSCLC cells [18], we attempted to determine whether autophagy exerts an influence on YZT-mediated mitochondrial apoptosis. A crucial process in mitochondrial apoptosis is the decrease in the mitochondrial membrane potential (MMP), which is subsequently followed by the release of mitochondrial cytochrome C to cytoplasm [19]. MMP can be measured by JC-1 staining assay in which the relative intensity of red/green fluorescence represents the ratio of mitochondrial depolarization. The results showed that combination with CQ intensified the YZT-mediated downregulation of MMP (Figure 5a). In addition, western blotting indicated that combination with CQ aggravated the YZT-mediated increase of Bax/Bcl-2 ratio and the level of cytochrome C (Figure 5b).

All these data suggest that YZT-induced autophagy plays a cytoprotective role partially by counteracting YZT-mediated mitochondrial apoptosis.

### 2.6. Compound YZT Initiates Autophagy through the PDK1/Akt Axis

Since YZT was synthesized by dual targeting of PDK1 and MEK, immunoblotting was applied to study the impact of YZT on the expression of PDK1. The results validated that YZT dose-dependently reduced the expression of p-PDK1(Ser241) in NSCLC cells (Figure 6a). Subsequently, we queried the-STRING-database (http://string-db.org, accessed on 28 September 2022) to analyze protein-protein interactions (PPIs) for PDK1. We observed that Akt and mTOR interact with PDK1 (Appendix A). Furthermore, Kyoto Encyclopedia of Genes and Genomes (KEGG) analysis indicated that PDK1 is an upstream protein of the Akt/mTOR axis (Appendix A). Increasing evidence suggests that the Akt/mTOR axis is a classical negative regulatory signaling route for autophagy [20,21], and we investigated whether YZT initiates autophagy via the Akt/mTOR axis. Immunoblotting demonstrated that YZT reduced the levels of p-Akt(Ser473), p-mTOR(S2448), p-4EBP1(Ser65) and p-p70s6k(Thours470) in a dose-dependent manner (Figure 6b). To validate whether YZT induces autophagy and regulates the Akt/mTOR axis via PDK1, p-Akt(Ser473) and LC3 were measured by immunoblotting in PDK1-overexprssing cells. As indicated in Figure 6c, overexpression of PDK1 partially eliminated the downregulatory effect of YZT on p-Akt(Ser473) and counteracted the YZT-mediated increase in LC3-II. To further study the impact of PDK1 on YZT-induced autophagy, immunofluorescence was used to monitor endogenous LC3 puncta when PDK1 was overexpressed. The results demonstrated that PDK1 overexpression significantly neutralized the YZT-mediated increase in endogenous LC3 puncta (Figure 6d).

In brief, YZT promotes autophagy by negatively regulating the PDK1/Akt axis in NSCLC cells.

### 2.7. Combined Administration with Chloroquine Strengthens the Antitumor Activities of Compound YZT In Vivo

A549 and H1299 cells were hypodermically injected into nude mice. When the xenotransplanted tumors became visible, the nude mice were allocated into three groups (5 mice/group) at random and injected with DMSO, YZT or YZT plus CQ through the tail vein. The dosage regimen was as follows: the dosages of YZT and CQ were 10 mg/kg and 80 mg/kg [9,22], respectively; the interval of administration was 2 days; and the total administration was six times (Figure 7a). Tumor volumes were measured before each drug administration. As indicated in Figure 7b, YZT markedly decreased the tumor volumes compared with the DMSO control. Furthermore, YZT plus CQ inhibited tumor growth more strongly than YZT alone. All images of tumor tissues were obtained on the 21st day after sacrifice (Figure 7c). In addition, immunohistochemistry demonstrated that the level of LC3 was increased by YZT (Figure 7d), which is consistent with the in vitro results.

The above results suggested that YZT inhibits NSCLC tumor growth in vivo and that combination with CQ can strengthen the anticancer activities of YZT in vivo.

## 3. Discussion

With the in-depth study of the biology of cancer and the development of proteomics, it has become gradually understood that tumorigenesis is jointly regulated by multiple signaling axes and multiple targeted proteins. Therefore, molecular targeted therapy has gradually replaced conventional cytotoxic medicines in the research and development of anticancer drugs. The RAF/MEK/ERK and PI3K/PDK1/AKT signaling axes exert critical influences in cancer and have become new targets for cancer therapy [23]. However, crosstalk and compensation mechanisms exist between these two pathways. Therefore, monotherapy targeting one of the two pathways might result in the acquisition of drug resistance in 6–8 months [24]. It was reported that the combination of MEK and PI3K inhibitors can suppress various tumors and is more effective than monotherapy [3,4,5,25]. Nevertheless, there might be two major problems with this combination: one is drug incompatibility and the other is the drug ratio, which should vary in different patients or even at different stages of cancer due to tumor heterogeneity. To solve the above obstacles, we designed a new prototype of MEK/PDK1 dual inhibitors, 3-substituted 5-phenylamin-indolone, by merging the core structural scaffolds of the MEK inhibitor PD0325901 and the PDK1 inhibitor BX517 under a fragment-based drug design strategy. Consequently, we obtained 43 new compounds with a common prototype structure, 3-substituted 5-phenylamin-indolone. In the new prototype structure, diphenylamine and indolone were designed to bind the allosteric pocket of MEK and reserve inhibitory activity against PDK1, respectively [6]. Our previous study revealed that Compound YZT significantly reduced the expression of MEK and PDK1, successfully achieving our design expectation [6]. This study further confirmed that YZT induced autophagy through negative regulation of the PDK1/Akt/mTOR axis. Considering that there are compensatory mechanisms in the RAF/MEK/ERK and PI3K/PDK1/Akt axes, our future research will clarify whether the novel MEK/PDK1 dual inhibitor YZT can overcome drug resistance.

Until now, the functions of autophagy have been classified into four types, cytoprotective, nonprotective, cytotoxic and cytostatic autophagy, in light of the roles of autophagy in cancer therapy. Among them, the characteristics of cytoprotective autophagy include improved therapy sensitivity and increased apoptosis when autophagy is blocked [26]. This study showed that blockade of autophagy by CQ or 3-MA could markedly strengthen YZT-mediated cytotoxicity and cell apoptosis, proving that YZT-induced autophagy is cytoprotective. Hence, the inhibition of cytoprotective autophagy can theoretically result in a more sensitive response to cancer treatment. The animal experiment in this study indicated that YZT significantly suppressed NSCLC tumor growth in vivo and that YZT plus CQ enhanced the antitumor activities compared with YZT alone. Our study showed that YZT is a promising novel drug candidate against NSCLC and that autophagy plays a pivotal role in the anticancer activities of YZT. To date, there are two main methods to block autophagy: pharmacological agents, i.e., autophagy inhibitors, such as CQ, 3-MA, wortmannin and bafilomycin A1, and genetic manipulation, i.e., via siAtg5 or siBeclin1 [27]. In this work, we preferred to employ the pharmacological agents CQ and 3-MA due to their convenience and feasibility for use in clinical practice.

Cell apoptosis generally involves two approaches: the death receptor pathway and the mitochondrial pathway (also called extrinsic and intrinsic apoptosis, respectively). The latter is featured by mitochondrial outer membrane permeabilization (MOMP), the reduction of mitochondrial membrane potential (MMP) and the discharge of mitochondrial cytochrome C to the cytoplasm [28]. Our previous study proved that YZT can induce mitochondrial apoptosis [18]. We wondered whether YZT-induced autophagy plays a cytoprotective role by intervening in mitochondrial apoptosis. In this study, compared to YZT alone, combination with CQ further lowered MMP, upregulated the ratio of Bax/Bcl-2 and the expression of cytochrome C, suggesting that YZT-induced autophagy exerts a cytoprotective role partially by antagonizing YZT-induced mitochondrial apoptosis. This conclusion is consistent with the reported findings that autophagy can control the kinetics of mitochondrial apoptosis and allow cellular recovery in apoptosis by regulating PUMA [29]. However, further research is needed to illuminate the detailed mechanisms underlying how YZT-induced autophagy intersects with mitochondrial apoptosis to tell a more complete story.

## 4. Materials and Methods

### 4.1. Cell Preparation

A549, H1299 and MRC-5 cells were maintained in DMEM and MEM, respectively, with 10% fetal bovine serum (Gibco, Waltham, MA, USA) and 10 U/mL penicillin-streptomycin in an incubator with 5% CO_2_ and 37 °C.

### 4.2. Antibodies and Reagents

P-PDK1(Ser241), Akt, p-Akt(Ser473), 4EBP1 and p-4EBP1(Ser65) and α-tubulin were purchased fromCell Signaling Technology (Danvers, MA, USA). P70S6K and p-p70S6K(Thr470) were acquired from Millipore (Boston, MA, USA). Antibodies against LC3, SQSTM1, Atg5, Beclin1, LAMP1, Bcl-2, Bax, cytochrome C, active caspase-9, active caspase-3, cleaved PARP, mTOR, p-mTOR(S2448) and β-actin were obtained from Abcam (Cambridge, UK). YZT was dissolved in DMSO and diluted with culture medium. CQ, 3-methyladenine (3-MA) and rapamycin (RAPA) were bought from Sigma-Aldrich (Saint Louis, MO, USA).

### 4.3. CCK-8

Appropriate cells were seeded into 96-well plates overnight, treated with drugs for 24 h, supplemented with 10 μL/well CCK-8 (Dojindo Laboratories, Kumamoto, Japan) and cultured for 2–4 h in an incubator. The OD values were detected under a microplate reader (Synergy HTX, BioTek, Burlington, VT, USA). 

### 4.4. BrdU Labeling

The procedures were implemented according to the instructions of the BrdU Cell Proliferation ELISA kit (Abcam, Cambridge, UK). The treatment times of drugs and BrdU were 24 h and 12 h, respectively. The OD values were determined under a microplate reader.

### 4.5. Colony Formation

Appropriate amounts of cells were seeded into 6-well plates and treated with YZT as indicated for 14 days. After fixation with 4% paraformaldehyde and staining with crystal violet, the cell clones were counted by a microscope. The cell aggregates were recognized as clones only when the cell numbers exceeded 50.

### 4.6. Annexin V-FITC/PI Apoptosis Detection

Following incubation with drug as indicated for 24 h, all cells, including the attached cells and the floating cells, were collected and stained with Annexin-FITC/PI (MultiSciences Biotech, Hangzhou, China). Apoptotic cells were determined with a flow cytometer (CyFlow Cube 6, Sysmex, Kobe, Japan).

### 4.7. Western Blotting

Approximately 10 μL of protein lysate was separated by 8–12% SDS-PAGE and transferred to PVDF membranes. After blocking with skim milk, the membranes were cut horizontally according to molecular weight of the tested proteins. The membrane strips were successively incubated with respective primary and second antibodies. The tested proteins were imaged by Immobilon Western Chemiluminescent HRP Substrate (Merck Millipore, Boston, MA, USA) on a scanner (Tanon 5500, Tanon Science & Technology, Shanghai, China). When the differences of molecular weight among the tested proteins were large enough, these proteins were run in the same SDS-PAGE gels; when molecular weight of the tested proteins was close/overlapping, these proteins were separated in different SDS-PAGE gels; when molecular weight of the tested proteins was close to that of β-actin, α-tubulin was selected as an endogenous control to normalize for the differences between the amount of total protein.

### 4.8. Immunofluorescence

The treated cells underwent fixation, permeabilization and blockade. After incubation with LC3 or LAMP1 overnight, goat anti-rabbit IgG H&L for 1 h and DAPI for 20 min, the cells were photographed under a laser scanning confocal microscope (FV1000, Olympus, Tokyo, Japan).

### 4.9. Transmission Electron Microscopy (TEM)

After fixation with 4% glutaraldehyde and 3% osmium tetroxide, dehydration by graded acetone and embedding in araldite, ultrathin sections were observed under a transmission electron microscope (7700, Hitachi, Tokyo, Japan). The autophagic bodies were counted with Image-Pro Plus 6.

### 4.10. Acridine Orange (AO) Staining

Cells were planted into 6-well plates overnight and treated with YZT at different concentrations for 24 h. After staining with AO solution (1 μg/mL) for 15 min and washing with PBS three times, the cells were measured by flow cytometry.

### 4.11. JC-1 Staining Assay

The operation was based on the instructions of the mitochondrial membrane potential (MMP) assay kit with JC-1 (Beyotime Biotechnology, Shanghai, China). The cells were stained with JC-1 staining solution, washed with JC-1 staining buffer solution and observed under a fluorescence microscope, taking the carbonylcyanide 3-chlorophenylhydrazone (CCCP) group as the positive control.

### 4.12. Overexpression Plasmid Transfection 

The HA-PDK1 overexpression plasmid of was acquired from GeneCopoeia (Rockville, MD, USA), transfected into the cells using Lipofectamine 2000 according to the instructions provided by the manufacturer and verified by western blotting.

### 4.13. Pharmacodynamics Experiment In Vivo

A total of 1 × 10^7^ A549 and H1299 cells were hypodermically implanted into nude mice aged between 6 and 8 weeks to establish two NSCLC tumor models. When the tumor volumes reached 50 cm^3^, the mice were assigned to 3 groups at random: DMSO, YZT and YZT plus CQ. The mice were administrated the drug every 3 days 6 times, and the tumor volumes were measured. On the 21st day, the tumor images were captured by a handheld imaging device (TM900, Peira, Beersel, Belgium) before sacrifice.

### 4.14. Immunohistochemistry

Briefly, after paraformaldehyde fixation and paraffin embedding of the tumor tissue, the sections were successively dewaxed, hydrated and treated with 0.01 M sodium citrate (pH 6.0), 3% hydrogen peroxide, and 1% goat serum with 0.2% Triton X-100. LC3 was used as the primary antibody. Subsequently, the slides were stained with HRP-conjugated goat anti-rabbit IgG H&L and visualized with 3, 3-diaminobenzidine. Photographs were taken under a microscope (IX50, Olympus, Tokyo, Japan).

### 4.15. Statistical Analysis

The data are presented as the mean ± SEM. Statistical analysis was carried out by SPSS Statistics 20. Statistical differences were determined by independent-samples *t* test or one-way ANOVA. Significant differences were designated as * *p* < 0.05, ** *p*< 0.01 and *** *p* < 0.001. All experiments were repeated at least three times.

## 5. Conclusions

This study shows that YZT possesses potent anticancer activities against NSCLC. Meanwhile, YZT promotes cytoprotective autophagy through the PDK1/Akt signaling axis. Furthermore, YZT-induced autophagy plays a cytoprotective role partially by counteracting YZT-mediated mitochondrial apoptosis. This study offers a reliable experimental support for developing YZT as an anti-NSCLC drug candidate and for the combined administration of YZT with CQ as a feasible therapy against NSCLC. 

## Figures and Tables

**Figure 1 pharmaceuticals-16-00244-f001:**
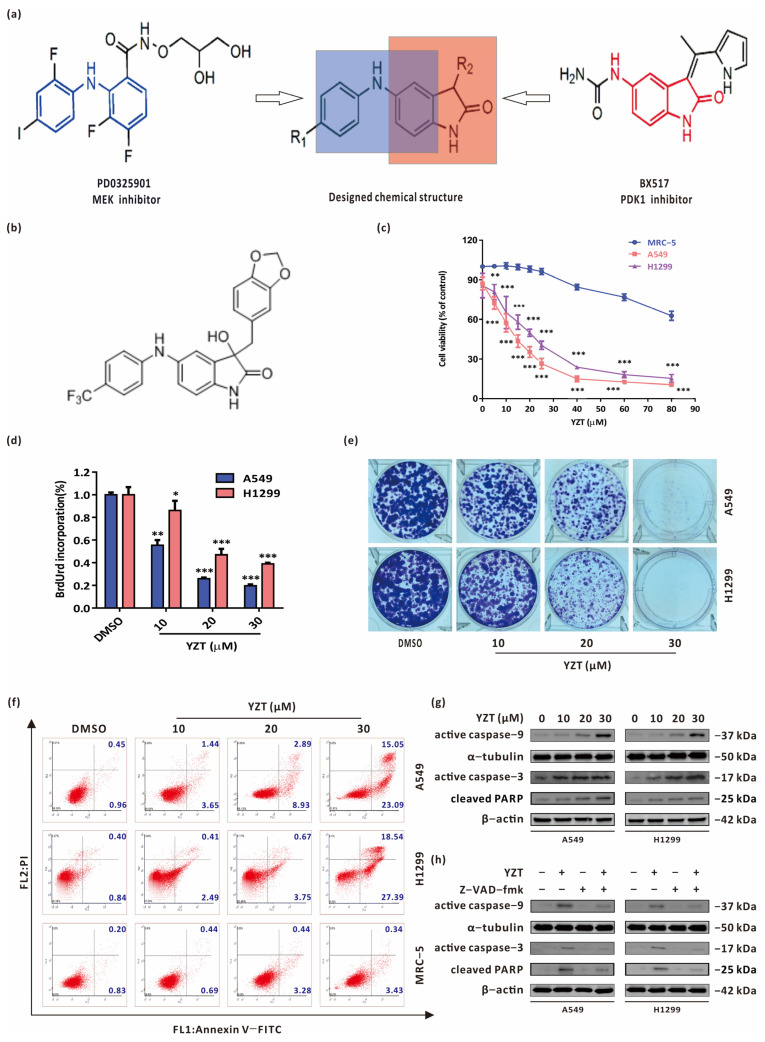
Compound YZT suppresses NSCLC cell growth. (**a**) Design and synthesis of the new MEK/PDK1 dual inhibitors. (**b**) Chemical structure of Compound YZT. (**c**) CCK-8 assay indicated that YZT dose-dependently reduced the viability of A549 and H1299 cells but has almost no influence on that of MRC-5 cells. (**d**) Cell proliferation was monitored by BrdU labeling detection. (**e**) Cell survival was measured by colony formation assay. (**f**) Annexin V-FITC/PI apoptosis assay revealed that YZT dose-dependently promotes apoptosis of A549 and H1299 cells but hardly has impact on that of MRC-5 cells. (**g**) Western blotting was used to analyze the expression of active caspase-9, active caspase-3 and cleaved PARP after YZT treatment. (**h**) NSCLC cells were pretreated with Z-VAD-fmk before YZT incubation, then Western blotting was applied to detect the expression of active caspase-9, active caspase-3 and cleaved PARP. Significant differences were designated as * *p* < 0.05, ** *p* < 0.01 and *** *p* < 0.001.

**Figure 2 pharmaceuticals-16-00244-f002:**
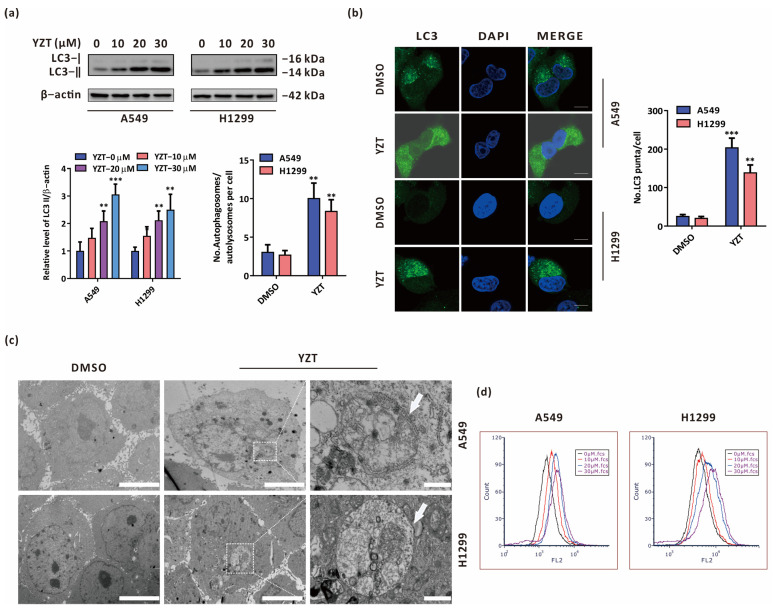
Compound YZT promotes autophagy in NSCLC cells. (**a**) The level of LC3 was determined by immunoblotting. (**b**) Endogenous LC3 puncta were monitored by immunofluorescence. Scale bar: 5 μm. (**c**) The formation of AVOs was detected by AO staining. (**d**) The ultrastructural morphology of autophagosomes/autolysosomes was visually observed under TEM. Scale bars: left and middle, 5 μm; right, 1 μm. Significant differences were designated as ** *p* < 0.01 and *** *p* < 0.001.

**Figure 3 pharmaceuticals-16-00244-f003:**
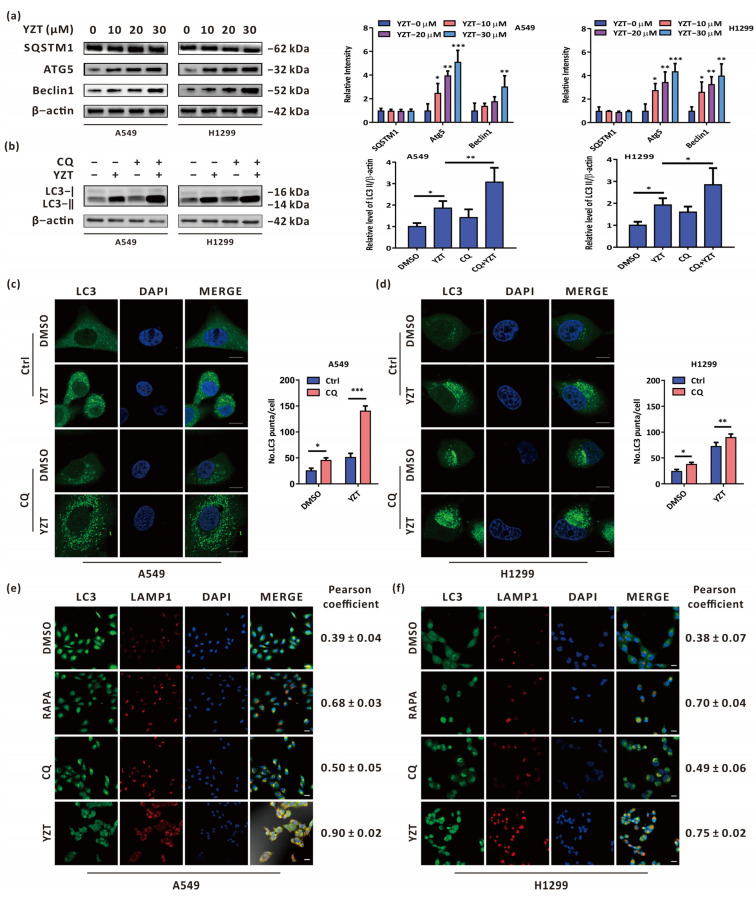
Compound YZT provokes complete autophagic flux. (**a**) The levels of SQSTM1, Atg5 and Beclin1 were analyzed by immunoblotting. (**b**) Combined treatment of YZT with CQ increases the level of LC3-II compared to YZT alone. (**c**,**d**) Endogenous LC3 puncta were monitored by immunofluorescence. Scale bar: 5 μm. (**e**,**f**) The colocalization of LC3 with LAMP1 was detected by a confocal microscope. Scale bar: 10 μm. Significant differences were designated as * *p* < 0.05, ** *p* < 0.01 and *** *p* < 0.001.

**Figure 4 pharmaceuticals-16-00244-f004:**
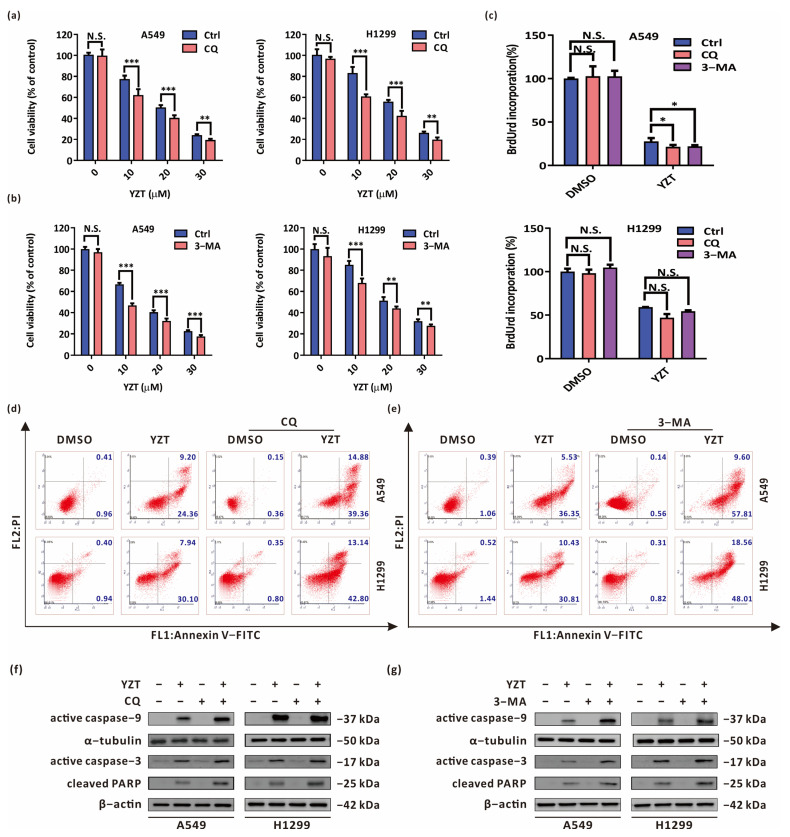
Autophagy blockade reinforces the anticancer activities of Compound YZT. (**a**,**b**) CCK-8 assay indicated that the combination of YZT with CQ (**a**)/3-MA (**b**) strengthens the inhibitory action on cell viability compared with YZT alone. (**c**) Cell proliferation was measured by BrdU labeling detection. (**d**,**e**) Annexin V-FITC/PI apoptosis assay indicated that combined administration of YZT with CQ (**d**)/3-MA (**e**) enhances cell apoptosis compared to YZT alone. (**f**,**g**) The expression of active caspase-9, active caspase-3 and cleaved PARP were examined by western blotting. Significant differences were designated as * *p* < 0.05, ** *p* < 0.01 and *** *p* < 0.001.

**Figure 5 pharmaceuticals-16-00244-f005:**
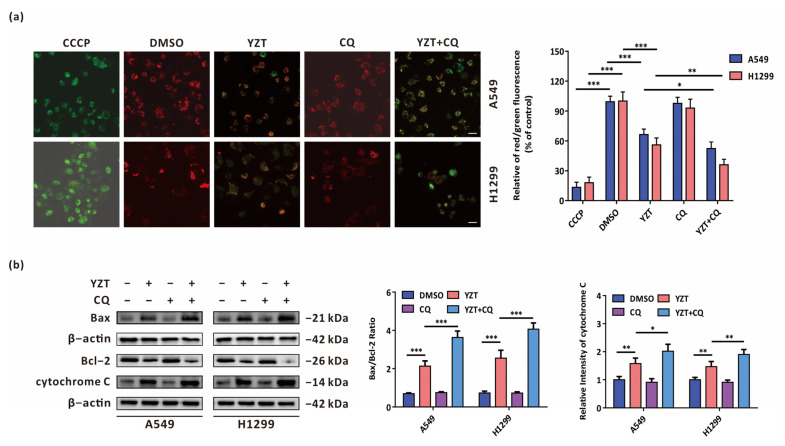
Autophagy inhibition aggravates YZT-mediated mitochondrial apoptosis. (**a**) JC-1 staining assay indicated that YZT plus CQ further lowers MMP compared with YZT alone. CCCP was used as the positive control. Scale bar: 10 μm. (**b**) Immunoblotting showed that combined treatment of YZT with CQ further upregulated the ratio of Bax/Bcl-2 and the level of cytochrome C compared to YZT alone. Significant differences were designated as * *p* < 0.05, ** *p* < 0.01 and *** *p* < 0.001.

**Figure 6 pharmaceuticals-16-00244-f006:**
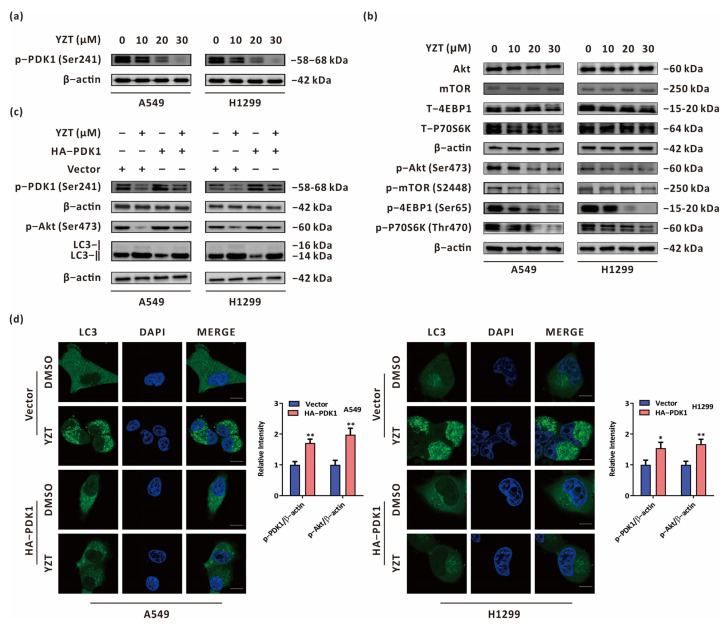
Compound YZT induces autophagy by negatively regulating the PDK1/Akt pathway. (**a**) The level of p-PDK1(Ser241) was detected by immunoblot. (**b**) The levels of p-Akt(Ser473), p-mTOR(S2448), p-4EBP1(Ser65) and p-p70S6K(Thr470) were examined by immunoblotting using the respective total protein levels as the internal controls. (**c**) Immunoblot analysis showed that PDK1 overexpression antagonized the YZT-mediated decrease in p-PDK1(Ser241), p-Akt(Ser473) and the increase in LC3-II. (**d**) Endogenous LC3 puncta were monitored by immunofluorescence. Scale bar: 5 μm. Significant differences were designated as * *p* < 0.05, ** *p* < 0.01.

**Figure 7 pharmaceuticals-16-00244-f007:**
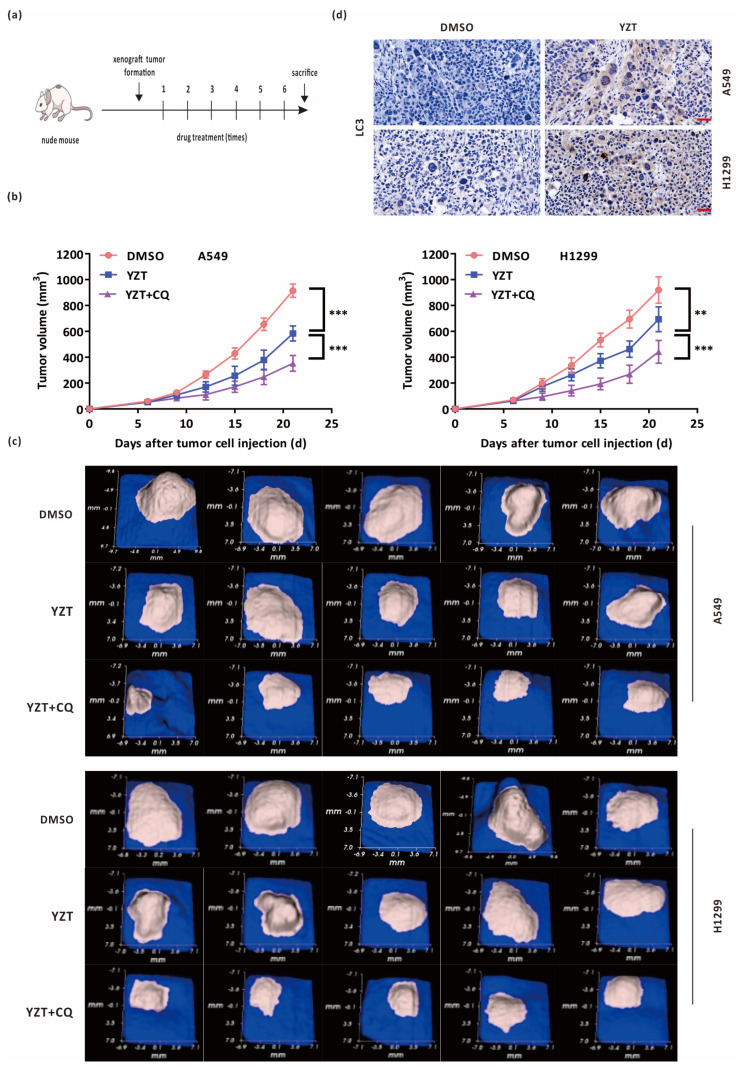
Combination with CQ intensified the antitumor activities of Compound YZT in vivo. (**a**) Time schedule of the animal study. (**b**) Tumor volumes at the different time points for the DMSO, YZT and YZT plus CQ groups. (**c**) Tumor images on the 21st day for the above 3 groups (5 mice/group). (**d**) Immunohistochemistry analysis of the levels of LC3 in the DMSO control and YZT groups. Scale bar: 40 μm. Significant differences were designated as ** *p* < 0.01 and *** *p* < 0.001.

## Data Availability

Data is contained within the article and Appendix A.

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
