# Peer review of "A Novel PDK1/MEK Dual Inhibitor Induces Cytoprotective Autophagy via the PDK1/Akt Signaling Pathway in Non-Small Cell Lung Cancer"

_pharmaceuticals, 2023, doi:10.3390/ph16020244_

Round 1

Reviewer 1 Report

Manuscript entitled “A novel PDK1/MEK dual inhibitor induces cytoprotective autophagy via the PDK1/Akt signaling pathway in non-small cell lung cancer” describes studies on the pro-apoptotic effect of YZT (compound previously described by the authors in Bioorganic & Medicinal Chemistry 27, 2019, 944-954) on NSCLC and a probable mechanism of anticancer activity. There are investigated pathways that induce autophagy in A549 and H1299 cell lines under influence of YZT alone as well as with additional autophagy inhibitors (chloroquine and 3-methyladenine).

The tested compound is known in literature and it should be referred in the Introduction. The main claims of the paper were properly placed in the context of the previous literature. The results of biological experiments support the claims and show that YZT promotes cytoprotective autophagy through the PDK1/Akt signaling axis. The title of the paper reflects the issues of presented work. The summary is too general and contains information in only five sentences. The language is adequate. The references are complete and recent. The Material and methods part is sufficiently described.

Reviewer recommends manuscript to further publishing process after major revision that is needed.

1.      The chemical name of YZT should be added to a main text

2.      There is lack of figure 1h caption

3.      Explain TEM abbreviation in section 2.2

4.      A photo in Figure 2c is illegible and should be enlarged

5.      A photo in Figure 6b is completely illegible incomprehensible to the average reader

6.      There are some mistakes such as 3-methyladenime (page 8, line 3 from the bottom, should be 3-methyladenine), cytochoursome (page 11, line 2, should be cytochrome).

7.      Conclusions should not contain Figure. Figure 8 can be a graphical abstract

Reviewer 2 Report

Dear authors,

I read your article with great interest, which I appreciate as an article of great scientific interest for the development of new compounds in the fight against cancer.

My recommendation will be to publish with minor revisions:

- I would like the authors to introduce a few extra phrases on how to design the compounds. The authors also refer to a database of compounds without attaching a bibliographic index (to the first discussion in the text about these compounds). How did the authors select the compound on which they later perform all the described tests. Is it the only dual inhibitor that showed antitumor properties?

- in figure 7, I recommend to the authors a clearer presentation of subsections c and d. In the legend d is immunohistochemistry.???

Round 2

Reviewer 1 Report

All comments were taken into consideration and the manuscript has been revised accordingly. I accept revised version.